# Morphology and Distribution of Antennal Sensilla in Three Species of Thripidae (Thysanoptera) Infesting Alfalfa *Medicago sativa*

**DOI:** 10.3390/insects12010081

**Published:** 2021-01-18

**Authors:** Yan-Qi Liu, Jin Li, Li-Ping Ban

**Affiliations:** College of Grassland Science and Technology, China Agricultural University, Beijing 100193, China; BS20203240989@cau.edu.cn (Y.-Q.L.); b20183040358@cau.edu.cn (J.L.)

**Keywords:** thrips, antennae, sensilla, SEM

## Abstract

**Simple Summary:**

To better understand how thrips detect and receive semiochemicals for intra- and interspecific communication, host selection, and mating, scanning electron microscopy was used to examine and compare the external morphology, distribution, and ultrastructure of the antennal sensilla in three alfalfa-feeding thrip species, *Odontothrips loti*, *Megalurothrips distalis*, and *Sericothrips kaszabi*. Nine major types of antennal sensilla were identified with similar morphology and distribution pattern among these thrip species. However, differences in the length and number of antennal sensilla were also found between male and female individuals and between species. Therefore, this study expands the understanding on the morphology of antennal sensilla in Thysanoptera (especially the suborder Terebrantia) and lays a morphological foundation for exploring the olfactory mechanism in thrips.

**Abstract:**

Thrips are important pests to alfalfa *Medicago sativa*. Similar as many other plant-feeding insects, thrips rely on the antennae to receive chemical signals in the environment to locate their hosts. Previous studies indicated that sensilla of different shapes on the surface of insect antenna play an important role in signal recognition. However, morphological analysis of the antennal sensilla in Thysanoptera has been limited to only a few species. To expand the understanding of how antennal sensilla are related to semiochemical detection in thrips, here we compared the morphology and distribution of antennal sensilla in three thrip species, *Odontothrips loti*, *Megalurothrips distalis*, and *Sericothrips kaszabi*, by scanning electron microscope (SEM). The antennae of these three species are all composed of eight segments and share similar types of sensilla which distribute similarly in each segment, despite that their numbers show sexual dimorphism. Specifically, nine major types of sensilla in total were found, including three types of sensilla basiconica (SBI, SBII, and SBIII), two types of sensilla chaetica (SChI and SChII), and one type for each of sensilla coeloconica (SCo), sensilla trichodea (ST), sensilla campaniformia (SCa), and sensilla cavity (SCav). The potential functions of sensilla were discussed according to the previous research results and will lay a morphological foundation for the study of the olfactory mechanism of three species of thrips.

## 1. Introduction

Alfalfa *Medicago sativa* is one of the most important forages worldwide cultivated, and thrips are one of the most destructive pests to alfalfa [1,2]. More than 10 thrip species, including *Odontothrips loti*, *Odontothrips confusus*, *Frankliniella occidentalis*, *Frankliniella intonsa*, *Thrips tabaci*, *Megalurothrips distalis*, and *Sericothrips kaszabi*, have also been documented as pests of legumes [1,3,4], and *O. loti* was the predominant one found in the legume-growing areas in China [5]. Thrips at their adult and nymph stages can damage the young tissues of alfalfa, including leaves and flower organs, resulting in leaf curling and withering. Studies showed that thrips could damage over 70% of plants in average (even as high as 100% in certain cases), seriously reducing the yield and quality of alfalfa hay [6].

Similar as many other herbivorous insects, thrips perceive chemical signals of plant volatiles and insect pheromones using their sensory organs for host recognition, feeding, and other life activities such as mating and oviposition [7,8,9,10,11]. For example, western flower thrips, *F. occidentalis* can be attracted by stereoisomers, a type of volatiles released by their host plants [12], and they can also react quickly when perceiving alarm pheromones released by other second-instar nymphs [13]. Another example is that male Poinsettia thrips, *Echinothrips americanus*, can avoid mating with mated females by identifying the pheromone dimethyl adipate (DBE-6) produced by males during mating [14].

The antennae are one of the principal sensory organs in insects for olfactory perception with various types of sensilla distributing along them [15,16,17,18,19]. Because sensilla act as important structures for gathering information from the environment and directing insects to respond properly, it is crucial to explore the antennal sensilla for understanding the mechanisms of host plant selection and feeding behaviors in insects.

At present, the type and the basic structure of antennal sensilla have been well documented in insects. Presumably, sensilla were classifiable according to their morphological structures or physiological functions. Sensilla chaetica, sensilla trichodea, sensilla basiconica, sensilla coeloconica, sensilla ampullacea, sensilla placodea, sensilla styloconica, sensilla squamiformia, sensilla campaniformia, sensilla scolopalia, and Böhm bristle are commonly classified according to structures [20,21]. Olfactory, tactile, and auditory receptors are examples of function-based terms; however, they may have also different structures. For instance, receptors associated with olfactory response are characterized by pores, e.g., sensilla basiconica, sensilla placodea in Coleoptera [16,22]. The pores of sensilla basiconica play an important role in stimulus conduction. Mechanoreceptors include sensilla chaetica, sensilla trichodea, sensilla campaniformia, etc. [23]. However, one sensillum may have more than one function in numerous arthropod species, e.g., sensilla trichodea, usually mechanoreceptors in Thysanoptera [24], may also have a sex pheromone-receptive function in Coleoptera [15], or olfactory chemoreceptor in some Hemiptera species [25].

Compared to most Lepidoptera and Coleoptera species, thrips are micro-insects with antenna of only about 300 nm long, difficult for research. Thus far, the ultrastructure of the antennae has only been demonstrated in a few thrip species [24,26,27,28,29,30,31], not including the three (*O. loti*, *M. distalis*, and *S. Kaszabi*) presented in this study. Despite that all these three are alfalfa feeders; *O. loti* is the major pest of alfalfa in China. In addition, with years of insecticides being applied to control thrip populations, the effects of pesticide residues and insecticide resistance are appearing [32], making it urgent to find sustainable thrip management approaches such as pheromone trapping [33], employment of insect-resistant plant varieties [34], and biological control [35]. Noteworthy, manipulation of olfactory behavior in some other thrip species was demonstrated to be effective in controlling their population, and several thrip-secreted olfactory chemicals have also been identified, including aggregation pheromone [33,36,37], alarm pheromone [13], contact pheromone [38], and anti-sex pheromone [14]. The trapping effect of aggregation pheromone on thrips is significant, and it has been widely used in the prediction and control of thrips [39,40]. In contrast, little was known on the olfactory mechanism of the three thrip species we investigated, and how the olfactory mechanism is related to the morphology of antennal sensilla in these species also remained unclear. Therefore, examination of antennal sensilla in these species may provide insights into the mechanism of their olfactory behavior and the development of a highly specific and sustainable approach for thrip management.

## 2. Materials and Methods

### 2.1. Insect Collections

The adult *O. loti*, *M. distalis*, and *S. kaszabi* used in this study were collected from alfalfa leaves at the west campus of China Agricultural University, Beijing, China (40°1′42″ N, 116°16′43″ E) in July 2018. Male and female *O. loti* were identified and separated under an anatomical microscope, and the samples were stored in 70% ethanol.

### 2.2. Scanning Electron Microscopy (SEM)

Ten insects for each gender were utilized for SEM observations. First, antennae were dissected from the heads, fixed in 70% ethanol for 2 h, and cleaned using an ultrasonic bath for 1–2 s each time by 3 times with ethanol (70%) replacement. The samples were then dehydrated in a series of ethanol solutions from low to high concentrations (60%, 70%, 80%, 90%, 95%, and 100%, respectively) with 2–3 min for each ethanol solution and were dried in a critical point dryer (LeicaEMCPD030, Wetzlar, Germany). Dried antennae were mounted on holder and gold-sputtered in a Hitachi sputtering ion exchanger (HITACHIMC1000, Tokyo, Japan). Lastly, the prepared samples were examined using a scanning electron microscope (HITACHISU8010, Tokyo, Japan) with the acceleration voltage set as 10 kV.

### 2.3. Data Analysis

The length and diameter of antennae and antennal sensilla in the three thrip species were measured from SEM images by using a biological microscope (i DM2300, Nangjing Jiangnan Yongxin Optical CO; LTD, China). Other information of antennal sensilla, including the number, distribution, and abundance of sensilla were also obtained from the SEM images. Data analysis and collation were completed in Statistical Product and Service Solutions 22.0. Images were annotated using Adobe Photoshop CC2018. Terminology usage and classification of antennal sensilla follow previous work by Snodgrass [20,41], Schneider [21], Zacharuk [42], Facci [31], Li [26], Zhu [28], Wang [27], and Hu [24].

## 3. Results

### 3.1. General Morphology of Antennae

In this study, the sensilla of *O. loti* were thoroughly examined in comparison with those of *M. distalis* and *S. kaszabi*. Antenna of three thrip species all consist of a scape, a pedicel, and a flagellum each, from the base to the tip. The flagellum is composed of six segments, named as FI, FII, FIII, FIV, FV, and FVI, respectively (Figure 1a,b). Nine major types of sensilla, including three types of sensilla basiconica (SBI, SBII, and SBIII), two types of sensilla chaetica (SChI and SChII), and one type of sensilla coeloconica (SCo), sensilla trichodea (ST), sensilla campaniformia (SCa), and sensilla cavity (SCav) each, were identified on the antenna surface with similar morphology and distribution pattern among these three thrip species (Figure 1c, Table 1 and Table 2).

Measurement of antennal length indicated that the three thrip species had antennae with length ranging from 261.27 μm to 355.12 μm (Appendix A). However, *O. loti* displayed sexual dimorphism in antennal length with 298.13 ± 3.27 μm in females and 261.27 ± 2.07 μm in males (Figure 2). Antennae of females were significantly (*p* < 0.05) longer than those of males.

### 3.2. Antennal Sensilla

#### 3.2.1. Sensilla Basiconica (SB)

SB was generally thick, with intensive longitudinal ridges and small holes on the surface. Three types of SB (SBI, SBII, and SBIII) were observed according to their morphological features.

SBI was distributed on the dorsal side of flagellum I and ventral side of flagellum II (Table 2). It located in sunken oval base, from which SBI bifurcates, forming a “V” or “U” shape. SBI surface was covered with shallow longitudinal ridges, and numerous pores were evenly distributed along the longitudinal ridges (Figure 3). The mean length of SBI in *O. loti* was 29.18 μm (29.57 ± 0.45 μm in females and 28.79 ± 0.70 μm in males, respectively). The mean length of SBI in *M. distalis* and *S. kaszabi* was 53.42 ± 1.81 μm and 24.87 ± 1.69 μm, respectively (Appendix A).

SBII was distributed on the outer ventro-lateral margin of flagellum II (Table 2), adjacent to SBI, short and thick, straight from the surface of antenna, and parallel to the axial direction of antenna. It was located in shallow pits than those on SBI. The base was as broad as the apex, and the surface was rough with longitudinal ridges and holes (Figure 4). In *O. loti*, the mean length of SBII was 6.86 ± 0.12 μm in females and 7.38 ± 0.32 μm in males. The mean length of SBII in *M. distalis* and *S. kaszabi* was 7.54 ± 0.30 μm and 8.35 ± 0.69 μm, respectively (Appendix A).

SBIII were distributed on flagellum III-V. Four SBIII were found on each antenna except that three were found on male *O. loti* (Table 1 and Table 2). The stout base of SBIII extends along the axial direction of the antenna, and the base to the apex is progressively thinned to form a cuspidal tip. The epidermis around the base is slightly sunken. The surface of SBIII is rough and evenly distributed with longitudinal ridges and holes (Figure 5). The length of SBIII distributed on the inner ventro-lateral margin of flagellum III was 8.88–13.11 μm in *O. loti*, 21.66–30.63 μm in *M. distalis* and 24.78–27.17 μm in *S. kaszabi*; The length of SBIII distributed on the inner ventro-lateral margin and the ventral side of flagellum IV was 15.98–31.67 μm in *O. loti*, 29.87–43.19 μm in *M. distalis* and 15.05–36.34 μm in *S. kaszabi*; The length of SBIII distributed on the dorsal side of the flagellum V was 18.82–21.14 μm in *O. loti*, 27.05–32.41 μm in *M. distalis* and 20.49–21.97 μm in *S. kaszabi* (Table 2).

#### 3.2.2. Sensilla Coeloconica (SCo)

SCo was located on the outer ventro-lateral margin of the flagellum III and flagellum IV (Table 2). It rises straight from the antennal surface and then bends to the apex of the antenna to be parallel to the antennal axis. The body is short and thick, with a smooth base and apex, separated by deeper arched ridges (Figure 6). The mean length of SCo in *O. loti* was 10.65 ± 0.90 μm in females and 8.47 ± 1.00 μm in males. The mean length of SCo in *M. distalis* and *S. kaszabi* was 10.89 ± 1.78 μm and 7.91 ± 0.85 μm, respectively (Appendix A).

#### 3.2.3. Sensilla Cavity (SCav)

SCav was located on the outer ventro-lateral margin of flagellum I and ventral side of flagellum III (Table 2). It is a circular opening brought by the invagination of the antennal cuticle (Figure 7). However, SCav was only found in both sexes of *O. loti* at the ventral side of flagellum III but were absent on the flagellum I. SCav is small with the average diameter for the three thrip species being approximately 500 nm.

#### 3.2.4. Sensilla Chaetica (SCh)

SCh was extensively distributed on all antennal segments. The length of SCh on each antennal segment varied greatly with the shortest one being only 11.94 μm and the longest being 53.74 μm (Table 2). SCh can be classified into two sub-types, SChI and SChII, according to the difference of its external morphology and distribution.

SChI was dispersed in scape, pedicel, and flagellum I-V (Table 2). On flagellum I-III, typically 4–5 SChI regularly surround the apex of the antennal segments. It was rooted in a shallow socket, protruding along the shaft of the antenna. The body slightly curved and was covered with longitudinal sharp edges without holes (Figure 8).

SChII’s distribution was confined on flagellum III-VI. The base of SChII was trapped in a slightly cuticular socket, and there were more arched ridges on the surface with blunt tips than those on SChI (Figure 9). A longitudinal opening in the sensillum made it distinguishable from other ridges (Figure 9c, shown by the black arrow). Moreover, secretions were observed to be attached to it, likely functioning for information exchange (Appendix A).

#### 3.2.5. Sensilla Campaniformia (SCa)

SCa was located at the dorsal end of the antennal pedicel (Table 2). It is formed by the protuberance of the round or oval cuticle, with a smooth surface and lying flat on the surface of the antenna (Figure 10). The mean diameter of SCa in *O. loti* was 5.56 ± 0.11 μm in females and 5.33 ± 0.61 μm in males. The mean diameter of SCa in *M. distalis* and *S. kaszabi* was 5.62 ± 0.58 μm and 4.81 ± 0.67 μm, respectively (Appendix A).

#### 3.2.6. Sensilla Trichodea (ST)

ST was situated on both sides of flagellum VI (Table 2). The slender body is curved in the apex. There is not any cuticular socket and the surface is smooth without holes (Figure 11). The mean length of ST in *O. loti* was 10.97 ± 0.19 μm in females and 8.27 ± 0.81 μm in males. The ST mean length in *M. distalis* and *S. kaszabi* was 15.53 ± 1.26 μm and 9.33 ± 0.99 μm, respectively (Appendix A).

### 3.3. Böhm Bristle (BB)

BB is rooted in the sunken basal fossa of the epidermis, perpendicular to the surface of the antenna, short, spiny, smooth without holes, and tapering from base to apex with slightly blunt tips (Appendix A). They are distributed in pairs or singly at the base of the scape or the intersegment joints between scape and pedicel on pedicel. The mean length of *O. loti* was 2.19 ± 0.14 μm in females and 1.64 ± 0.16 μm in males. The BB mean length in *M. distalis* and *S. kaszabi* was 2.56 ± 0.51 μm and 3.35 ± 0.54 μm, respectively.

### 3.4. Microtrichia (Mt)

Microtrichia is generally distributed in flagellum I-IV. A single sensillum is short and smooth with cuspate tips (Figure 1a,b and Appendix A). They arrange transversely from the base in a circle, with each antennal segment being surrounded by 3–4 circles. In addition, microtrichia were found on the dorsal side of the *S. kaszabi* pedicel with hierarchical arrangement but not on the ventral side.

## 4. Discussion

For most phytophagous insects, semiochemicals in the environment, host plant volatiles and sex pheromones, for example, are crucial for their survival and reproduction, because by perceiving these signals, the insects are able to find their host plants for feeding and the same species but opposite-sex of insect individuals for mating. Antennal sensilla act as external “receivers” to accurately receive environmental signals from a variety of distances, then triggering a complexity of behaviors as response. The morphology of the receptors at different developmental stages in Thysanoptera have been reported previously [24,26,27,29,30,43], but their functions were mostly unknown [28,31]. Noteworthy, a high degree of morphological and positional similarities were found in the antennal sensilla of Thripidae, making it reasonable to hypothesize that sensilla sharing similar morphological characteristics may execute same functions in signal perception [44,45].

In this study SB was the only multiporous sensilla observed, which appeared to be the homologous tissues of the multiporous homonymic sensilla, has been described by Zhu in chilli thrips, *Scirtothrips dorsalis* [28], Hu in *E. americanus* [24], and Li in *F. intonsa, Frankliniella tenuicornis*, and *F. occidentalis* [26]. These sensilla are innervated by highly branched dendrites and are connected to the outside through pores [28]. This porous structure provides the possibility for a variety of olfactory proteins in lymph to bind to odor molecules in the air [46], and it is considered to be a receptor closely related to olfactory response, e.g., sensilla basiconica are activated by 4-Vinylanisole, which is an aggregation pheromone, and play a role in locust aggregation behavior [47]. In addition, sensilla basiconica responds to plant volatiles in *Phoracantha semipunctata* [22], *Cactoblastis cactorum* [48], *Adelphocoris Lineolatus* [49], and *Drosophila* [50].

We also found that SCo is a peg-like tissue rooted in the shallow fossa. A similar sensillium was previously reported on the antenna of *S. dorsalis* [28]. Examination on the fine structure of SCo in chilli thrips revealed it as a double-walled tissue. Various spoke channels lead the inner cavity to the groove channels, and an inner cuticular wall surrounded unbranched dendrites. Based on these characteristics and the previous reports, SCo may function as olfactory receptors [28,31].

SCh is a major antennal sensillum found in the three thrip species investigated here as they were extensively distributed along all antennal segments. Regarding to its function, two alternative hypotheses were proposed: it functions merely as a mechanoreceptor [23], or it has dual functions as a mechanoreceptor and as a contact chemosensitive receptor because it can be innervated by both mechanosensitive neurons and chemosensitive neurons [31]. In this study, we found that SChII was connected with a longitudinal opening, which was consistent with the single hole described by Zhu [28]. Meanwhile, the secretion attached to the opening was observed (Appendix A) and we therefore speculated that SChII located at the end of the antenna had mechanical sensing function and played a role in communicating with the environment.

ST has a slender and smooth body, which was located on the distal part of the antenna. A sensillum, termed “tactile hair”, was previously reported for perceiving the mechanical distortion of the body, caused by external stimuli or the internal forces generated by activities of the muscles [23,43]. However, in Thysanoptera, no internal ultrastructure was found in ST, and therefore we suspected that it is a mechanoreceptor based on its positions and previous reports [28].

SCa exists on the dorsal side of the pedicel in Thysanoptera [51], which was observed starting from the nymph stage [52]. Its long existence and conservation in locations and numbers suggested that SCa is functionally conserved, possibly playing a role in sensing and responding to the tension produced by insect exoskeletons [23].

SCav is a cavity-like sensillum with an inward depression in the stratum corneum and often adjacent to SB or SCo. It follows the distribution law of Thermo-hygroreceptors and further confirmed in the ultrastructure of *S. dorsalis* [28,45]. In this study, the similar sensilla were classified as sensilla cavity and likely functions in the perception of environmental humidity and temperature changes.

BB exists conservatively in the scape and pedicel of the antenna of many species of insects [15,53,54,55]. It is generally regarded as a mechanoreceptor that senses tension and controls the antenna to swing within a safe range [21]. However, in Thysanoptera, the ultrastructure of Böhm bristle is not reported, and its function has not been verified [28]. Therefore, it is not defined as a functional receptor in this paper. Microtrichia was the most abundant structures on the antenna, also acting as a derivative structure of the cuticular widely distributed on flagellum I-IV [24,27,28].

In this study, nine major types of sensilla in total were identified on the antenna with similar morphology and distribution pattern among three species of thrips. From the phylogenetic relationship, *Odontothrips* and *Megalurothrips* are sister taxa, while *Sericothrips* belongs to Sericothripinae, which is a monophyletic group [56]. This explains that the morphological difference of sensilla between *O. loti* and *M. distalis* is small, but sensilla of *S. kaszabi* are quite different from those of the other two thrip species. Sex differentiation may be related to the function of sensilla, which needs to be further verified. A division in subtypes was difficult given the lack of further functional verification results, such as electroantennogram, and behavior assay. Consequently, the classification and function need to be clarified by further study in Thysanoptera. Furthermore, the screening of olfactory-related proteins and the separation and identification of pheromones are being carried out to further do more in-depth research on the olfactory mechanism of thrips on the basis of clarifying the type of antennal sensilla.

## 5. Conclusions

This study provided the complete antennal morphology of the three thrip species parasitic on alfalfa (*O. loti*, *M. distalis*, and *S. kaszabi*), with emphasis on the types, number, and location of sensilla. The results show that, nine major types of sensilla in total were identified on the antenna with similar morphology and distribution pattern among three species of thrips, which included three types of sensilla basiconica (SBI, SBII, and SBIII), two types of sensilla chaetica (SChI and SChII), and one type for each of sensilla coeloconica (SCo), sensilla trichodea (ST), sensilla campaniformia (SCa), and sensilla cavity (SCav). However, there are different in the length and number of antennal sensilla between male and female individuals and between species.

A large number of morphological evidences show that morphologically similar sensilla have the same function [31,45], which advances the study of morphology. Among different species of Thripidae, antennal sensilla have high similarity in types and position, suggesting that they may have similar functional responses. However, further verification of the function of the sensilla also needs to be combined with electrophysiological and behavioral responses. This study is of certain significance to supplement the morphological study of antennal sensilla of Thysanoptera (especially Terebrantia) and lays a morphological foundation for the study of the olfactory mechanism of three species of thrips.

## Figures and Tables

**Figure 1 insects-12-00081-f001:**
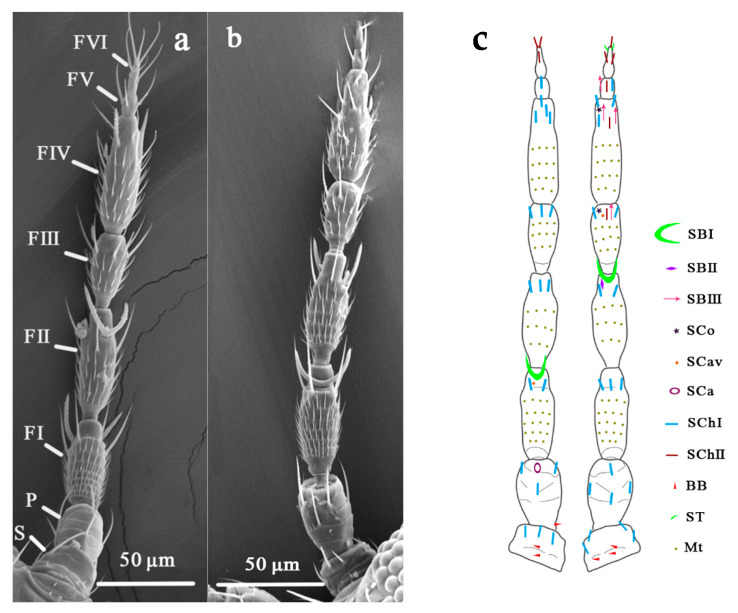
An overview of the general morphology and sensilla distribution on the thrip antennae. (**a**) ventral view of an *O. loti* antenna under SEM; (**b**) dorsal view of an *O. loti* antenna under SEM; (**c**) the schematic distribution of various types of sensilla on the dorsal (left) and ventral (right) side of three thrip species investigated. S: Scape. P: Pedicel. FI-VI: Flagellum I-VI. SBI-III: sensilla basiconica I-III. SCo: sensilla coeloconica. SCav: sensilla cavity. SCa: sensilla campaniformia. SChI, II: sensilla chaetica I, II. ST: sensilla trichodea. BB: Böhm bristle. Mt: microtrichia.

**Figure 2 insects-12-00081-f002:**
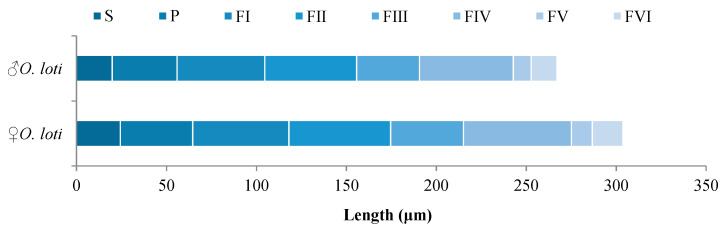
Average length (n = 10) of each antenna segment in *O. loti*. S: Scape. P: Pedicel. FI-VI: Flagellum I-VI.

**Figure 3 insects-12-00081-f003:**
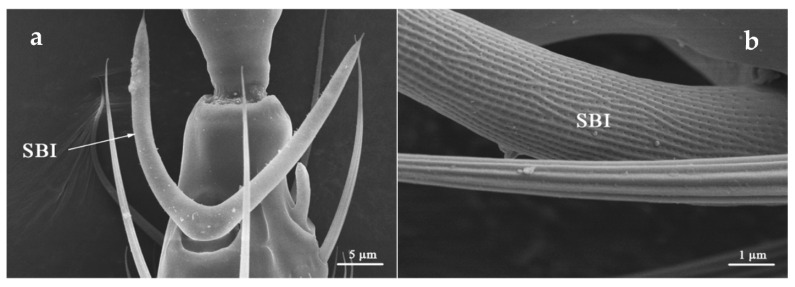
Sensilla basiconica I (SBI) is located at the flagellum II apex with holes on its surface. SBI of *O. loti* (**a**) and *S. kaszabi* (**b**) is shown.

**Figure 4 insects-12-00081-f004:**
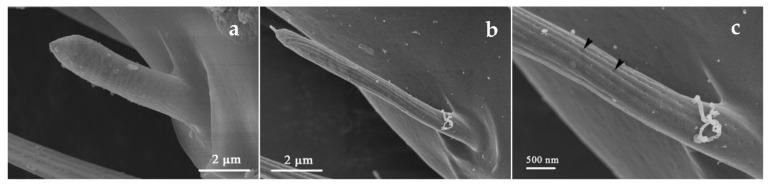
Sensilla basiconica II (SBII) on the flagellum II of *O. loti* (**a**) and *S. kaszabi* (**b**). A high-resolution of SBII is shown in (**c**). Holes (indicated by the black arrows) are distributed evenly between longitudinal ridges but with a smaller density than that of sensilla basiconica I (SBI).

**Figure 5 insects-12-00081-f005:**
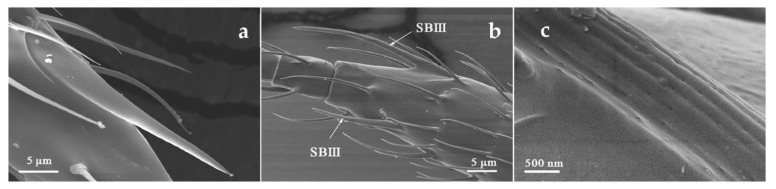
Sensilla basiconica III (SBIII) on the flagellum IV of *O. loti* (**a**), *S. kaszabi* (**b**), and SBIII (**c**) shown in high resolution.

**Figure 6 insects-12-00081-f006:**
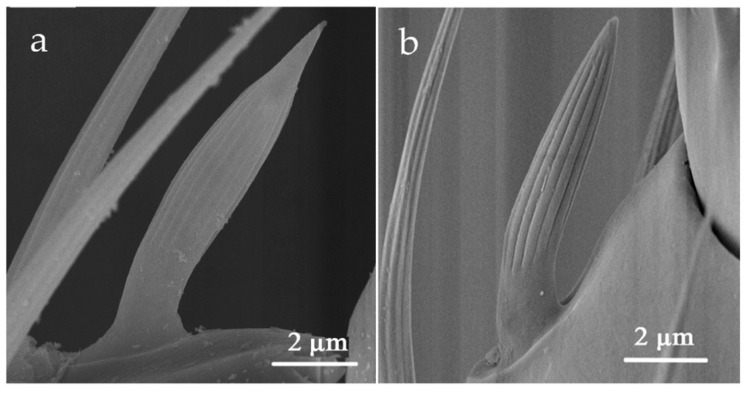
Sensilla coeloconica (SCo) on the flagellum III in *O. loti* (**a**) and *S. kaszabi* (**b**).

**Figure 7 insects-12-00081-f007:**
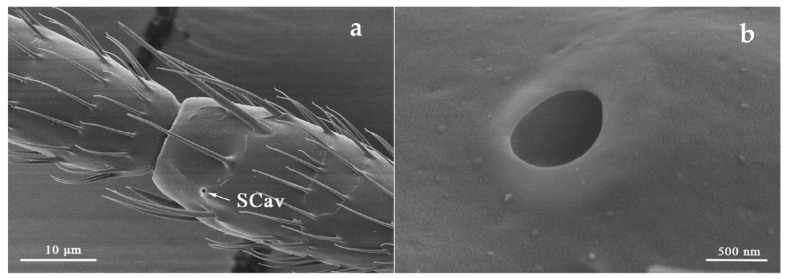
Sensilla cavity (SCav) is located on flagellum III (**a**) and a SCav was shown in higher resolution (**b**) in *S. kaszabi*.

**Figure 8 insects-12-00081-f008:**
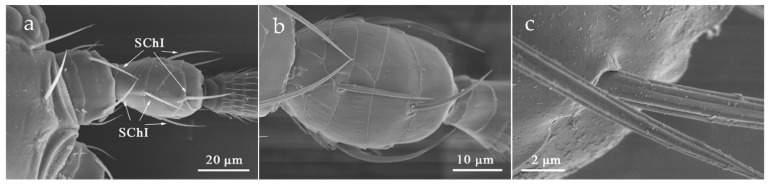
Sensilla chaetica I (SChI) on the scape and pedicel. (**a**) *O. loti*. (**b**) *S. kaszabi*. (**c**) *M. distalis*.

**Figure 9 insects-12-00081-f009:**
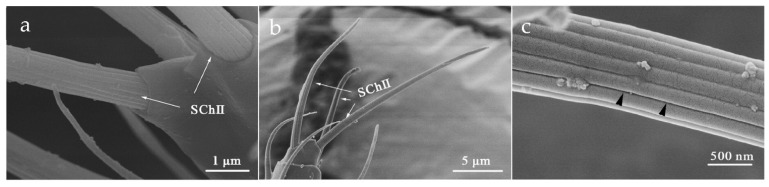
Sensilla chaetica II (SChII) on the flagellum VI. (**a**) *O. loti*. (**b**) *S. kaszabi*. (**c**) *M. distalis*. There is a longitudinal opening at the SChII indicated by the black arrow.

**Figure 10 insects-12-00081-f010:**
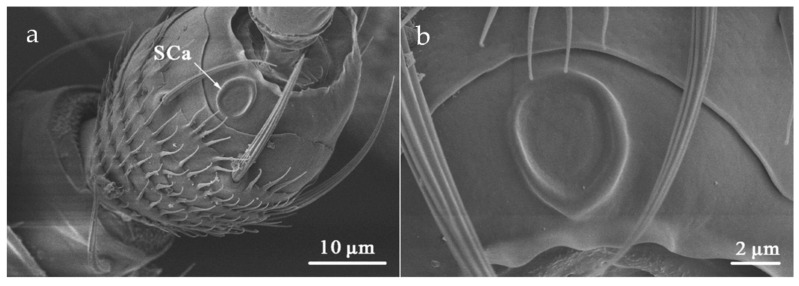
Sensilla campaniformia (SCa) on the pedicel (**a**) and shown at higher magnification (**b**) in *S. kaszabi*.

**Figure 11 insects-12-00081-f011:**
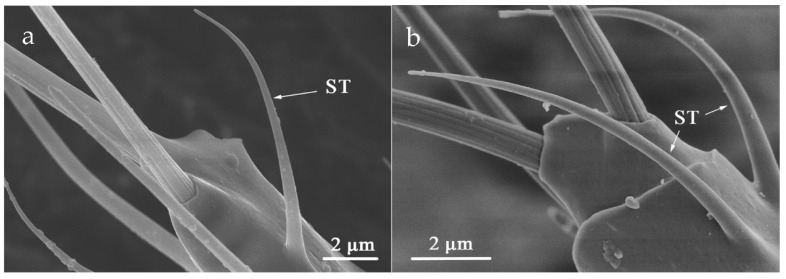
Sensilla trichodea (ST) at the flagellum VI. (**a**) *O. loti*. (**b**) *S. kaszabi*.

**Table 1 insects-12-00081-t001:** The number of nine different types of sensilla in three thrip species.

Sensilla Type	*Odontothrips loti* (n = 5)	*Megalurothrips distalis* (n = 5)	*Sericothrips kaszabi* (n = 5)
s. basiconica I	2	2	2
s. basiconica II	1	1	1
s. basiconica III	3 (4) *	4	4
s. cavity	1	2	2
s. coeloconica	2	2	2
s. chaetica I	38	38	35
s. chaetica II	9	9	9
s. campaniformia	1	1	1
s. trichodea	2	2	2

*****: The number of sensilla basiconica III is 3 in males and 4 in females.

**Table 2 insects-12-00081-t002:** The number, location, and length of antennal sensilla in three thrip species.

Antennal Segment	Sensilla	The Number	Position	Length (μm)		
				*Odontothrips loti* (n = 5)	*Megalurothrips distalis* (n = 5)	*Sericothrips kaszabi* (n = 5)
Scape	s. chaetica I	5 (7) *	verticillate	11.94–28.38	17.89–29.76	17.10–28.68
Pedicel	s. campaniformia	1	dorsal	3.42–3.81	3.34–5.62	4.05–5.28
	s. chaetica I	6 (7)	verticillate	16.38–23.70	20.74–33.65	16.09–26.23
Flagellum I	s. basiconica I	1	dorsal	22.28–31.65	52.61–55.56	23.35–25.46
	s. cavity	1 (0)	outer ventro-lateral	--	0.31–0.66	0.61–0.71
	s. chaetica I	5	verticillate	20.38–29.52	37.05–53.74	15.72–25.06
Flagellum II	s. basiconica I	1	ventral	23.85–31.50	51.39–54.14	22.35–25.99
	s. basiconica II	1	outer ventro-lateral	6.25–9.10	7.33–7.76	7.58–8.92
	s. chaetica I	5	verticillate	19.75–29.73	28.51–41.68	13.04–25.39
Flagellum III	s. basiconica III	1(0)	inner ventro-lateral	8.88–13.11	21.66–30.63	24.78–27.17
	s. coeloconica	1	outer ventro-lateral	7.40–11.06	12.14–12.86	7.22–8.87
	s. cavity	1	ventral	0.45–0.75	0.46–0.48	0.55–0.83
	s. chaetica I	5	verticillate	14.85–23.97	27.88–33.57	12.55–19.57
	s. chaetica II	1	ventral	14.96–22.57	23.19–27.92	15.01–18.67
Flagellum IV	s. basiconica III	2	inner ventro-lateral/ventral	15.98–31.67	29.87–43.19	15.05–36.34
	s. coeloconica	1	outer ventro-lateral	6.89–11.40	8.70–9.76	6.61–8.50
	s. chaetica I	7	verticillate	12.07–21.62	23.35–27.49	12.57–18.98
	s. chaetica II	1	ventral	17.89–22.49	21.34–29.04	13.75–16.58
Flagellum V	s. basiconica III	1	dorsal	18.82–21.14	27.05–32.41	20.49–21.97
	s. chaetica I	2	inner ventro-lateral/dorsal	13.13–19.76	19.49–26.26	14.34–15.55
	s. chaetica II	1	ventral	13.75–19.35	19.24–26.40	14.54–16.98
Flagellum VI	s. chaetica II	6	verticillate	15.11–22.59	23.08–34.91	11.20–19.67
	s. trichodea	2	lateral	7.18–11.17	13.84–17.25	7.69–10.17

**Note:** The values in the table represent the range from the minimum to the maximum length. S. cavity, S. campaniformia measures the range of diameters. *: The numbers in parentheses represent the variation number of intra- and interspecific sensilla.

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
