# Peer review of "Morphology and Distribution of Antennal Sensilla in Three Species of Thripidae (Thysanoptera) Infesting Alfalfa Medicago sativa"

_insects, 2021, doi:10.3390/insects12010081_

Round 1

Reviewer 1 Report

INSECTS – 1029665 _ A Review

The authors claim to have examined the surface structure of three named species of Thripidae, and that Prof Feng Jinian was responsible for the insect identifications. However there has clearly been some error in preparing the specimens for study, because the images provided do not seem to display the species named. Figure 1 and Figure 5a are not of Odontothrips loti – that species is well known for the greatly enlarged SBIII on the outer ventro-lateral margin of flagellum IV. In loti the sensory structure in that position occupies much of the surface of the antennal segment. Similarly, Figure 5b does not seem to be of Sericothrips kaszabi, because in that species the base of SBIII on the inner ventro-lateral margin of flagellum IV is long and very narrow and 50% of the length of the segment.

Certainly, the authors have displayed their technical expertise in producing some nice SEM images. However, the significance of their work is far from clear. They admit that the types and distribution of antennal sensilla are similar amongst relatively unrelated species of Thripidae, so the observations given here are not particularly original. The suggested significance of this work to the control of thrips pests is even more difficult to understand. I am not convinced that this mss is suitable for publication in Insects.

Reviewer 2 Report

This article describes the antennal sensillae of thrips, tiny pest insects. Obtaining information on their biology is useful in the context of sustainable strategies of pest managements.

As English is not my first language, I usually refrain from making language comments; but here, I believe many sentences need revisions.

While the work is clear, I think some details could be added in the results or the discussion. No comment is made on the difference between males and females for the number of sensilla basiconica III. As they are olfactory, could this be related to sex-pheromone perception? I do not expect a long discussion, but only one or two sentences to highlight this. Similarly, sometimes the numbers of sensillae or their sizes are different between the three species: I would have expected some statistics here for the sizes (e.g. l. 122-124), solely to tell the reader whether the observed variations reflect biological differences or random variations. I realize that it is not possible to provide a functional analysis, but it would be useful to report whether or not the three species have a similar ecology or what are their phylogenetic relationship, and to make the connexion with these differences. Alternatively, if these data are is not known, this should be stated.

Why not commenting on Böhm bristle and microtrichia?

In table 2, some figures are in parenthesis, why? This should be explained.

For the insect collection: beyond the sex, it would be useful either know the developmental stage (as they are heterometabolous, the collected specimens might be adult or young) or to specify that this could not be observed. This is not a problem, but this needs to be specified. Similarly, the month/season of collection should be reported.

  1. 302: “Please add” should be removed.

Round 2

Reviewer 1 Report

Lines 308-309 are clearly incorrect. They are based on an assumption that if a thrips is found on a plant, then that thrips is feeding on and damaging that plant. There is NO evidence that any Megalurothrips species feeds on any plants other than species of Fabaceae (=Leguminosae). Similarly, I doubt that S. kaszabi "is harmful to many kinds of weeds". These comments have no experimental basis, and since lines 308-309 are tangential to the careful work of these authors I suggest deleting these lines. 

Author Response

Response 1: Thank you for your valuable advice. Lines 307-311 have been deleted.
